# Enhancing Neurofibroma Segmentation in Whole-Body MRI: Leveraging an Anatomy-Informed Approach

**Georgii Kolokolnikov**[1,2]                                    G.KOLOKOLNIKOV@UKE.DE
**Marie-Lena Schmalhofer**[3]                                    M.SCHMALHOFER@UKE.DE
**Inka Ristow**[3]                                                I.RISTOW@UKE.DE
**René Werner**[1,2]                                              R.WERNER@UKE.DE

[1] *Institute for Applied Medical Informatics, University Medical Center Hamburg-Eppendorf*

[2] *Institute for Computational Neuroscience, University Medical Center Hamburg-Eppendorf*

[3] *Department of Diagnostic and Interventional Radiology and Nuclear Medicine, University Medical Center Hamburg-Eppendorf*

**Editors:** Accepted for publication at MIDL 2024

## Abstract

This study presents an anatomy-informed segmentation approach for neurofibroma in fat-suppressed T2-weighted whole-body MRI (WB-MRI). By adapting TotalSegmentator for WB-MRI segmentation and employing dedicated Dynamic UNet models across four anatomical zones, we achieved improvements of 20% in terms of the Dice coefficient on a test set. The proposed method promises to streamline neurofibroma segmentation, emphasizing future integration into interactive workflows.

**Keywords:** Neurofibroma, whole-body MRI, deep learning, medical image segmentation, anatomy-informed approach, TotalSegmentator.

## 1. Introduction

Neurofibromas (NFs), associated with the genetic disorder neurofibromatosis type 1 (NF1), manifest as tumors along nerves in the skin and deeper soft tissues (Lammert et al., 2005). Monitoring NF1 patients is critical due to the potential for malignancy with a high tumor burden being a risk factor for malignant transformation (Korf, 1999). Whole-body MRI (WB-MRI) is an important clinical feature in NF diagnosis (Legius et al., 2021), yet challenges in manual tumor quantification (Kollmann et al., 2020) and the subjectiveness of human observers prompt a need for enhanced methods. NF segmentation, vital for the analysis, is time-intensive and challenging, with manual segmentation taking up to five hours per patient and state-of-the-art automatic methods like nnU-Net showing only 25% accuracy (Zhang et al., 2022). While semi-automatic interactive methods (Diaz-Pinto et al., 2022) improve accuracy, they still demand significant manual intervention in case of high tumor load. Improved automatic pre-segmentation in addition to the interactive pipeline alleviates the manual effort.

We propose an anatomy-informed approach to the automatic pre-segmentation of NF in WB-MRI. As it was reported in (Zhang et al., 2022), a default nnU-Net pipeline shows a low performance on this task. Hence, we leverage the acquisition of anatomical information from accompanying T1-weighted (T1w) data. Our contributions are: 1) establishing a pipeline for anatomy segmentation based on the adaptation of the TotalSegmentator; 2) inclusion of the anatomical information in the NF segmentation; 3) anatomical zone based NF segmentation.

## 2. Materials and Methods

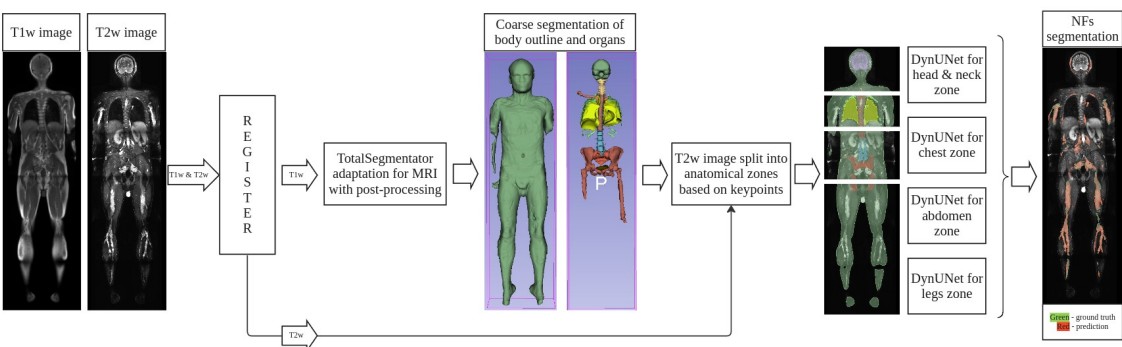

Figure 1: Anatomy-informed neurofibromas (NFs) segmentation pipeline. T1w and T2w MRIs are registered. T1w is segmented with the adapted TotalSegmentator to get anatomical masks. T2w is split into anatomical zones. Each zone is segmented with a dedicated DynUnet to get respective NFs masks. The NFs masks are stitched together to form the final segmentation mask.

**Dataset**. Single-center data was acquired at the University Medical Center Hamburg-Eppendorf. The dataset contained T1w and T2-weighted (T2w) fat-suppressed WB-MRIs of 60 patients covering a period of 14 years (2006 - 2020). NF tumors in the dataset were manually segmented with ITK-SNAP 3.8.0 (Yushkevich et al., 2006) by two radiologists (I.R. and M.-L. S.). The WB-MRIs were randomly split on a patient level into train (48 patients, 70 MRIs, median tumor burden 319 ml), validation (6 patients, 12 MRIs, median tumor burden 413 ml), and test subset (6 patients, 11 MRIs, median tumor burden 602 ml). The dataset underwent spacing alignment, Gaussian smoothing, and intensity scaling.

**Anatomy-informed segmentation**. NFs show a predisposition for the head/neck, trunk, and extremities locations (Staser et al., 2012). According to the initial data analysis, the appearance of NFs depends on their localization within a body. For example, NFs located in the legs can look similar to blood vessels, and those in the neck can resemble lymph nodes. Given varied appearances of NFs across anatomical zones (Staser et al., 2012), providing location of organs and considering the zones separately could enhance segmentation accuracy. We used T1w WB-MRI to capture anatomical structures, since it provides high contrast between fat and water-containing tissues. For identifying NFs, we used fat-suppressed T2w WB-MRI, which is more sensitive to a fluid content (Ahlawat et al., 2016). T1w and T2w WB-MRIs were rigidly registered to compensate for a patient positioning (Figure 1). We trained the TotalSegmentator (Wasserthal et al., 2023) using an unsupervised domain adaptation method to adapt it to the highly-anisotropic T1w WB-MRI data following (Weihsbach et al., 2023). The adapted model segmented key anatomical features in T1w WB-MRI, including body surface and internal organs, which were refined by removing inaccuracies through connected component analysis. The T2w WB-MRI was divided into four anatomical zones (head-neck, chest, abdomen, and legs) based on the landmarks identified from the anatomy segmentation. By dividing the body into these

zones, we tailored the analysis to each region, accommodating the appearance of tumors in different body parts. We trained four Dynamic UNets (DynUNets) with anisotropic kernels and strides suited for each zone. The segmentation was performed patch-wise. An anatomy segmentation mask was passed as the second channel for each DynUNet. Segmentations of NF for each zone were stitched together to form the whole-body NF mask. Since the assessment of internal tumor load is of major importance for physicians, we applied masking with a morphologically eroded body outline to focus only on the deep soft tissue NFs and to exclude superficial cutaneous NF that do not tend to undergo malignant transformations.

## 3. Results

To evaluate the effect of the anatomy-informed approach (Table 1), we compared the baseline method, whole-body NF segmentation with DynUNet (patch size 128x128x32) without any anatomical information included (denoted as WB), the WB method with anatomy masks given as the second channel (WBA), an anatomical zone based segmentation with a set of DynUNet models described above (ZB), and the ZB method with anatomy masks given as the second channel (ZBA). Additionally, we checked the effect of masking the segmentation with an eroded body outline (removed 1 cm below the skin surface). Two metrics were used: Dice similarity coefficient (DSC) and volume overlap error (VOE).

Table 1: Comparison of the performance of the segmentation pipelines.

| Segmentation | Validation DSC ($\uparrow$) | Validation VOE ($\downarrow$) | Test DSC ($\uparrow$) | Test VOE ($\downarrow$) |
|---|---|---|---|---|
| WB (baseline) | $0.34 \pm 0.19$ | $0.78 \pm 0.15$ | $0.32 \pm 0.08$ | $0.79 \pm 0.07$ |
| WB + masking | $0.34 \pm 0.19$ | $0.77 \pm 0.15$ | $0.35 \pm 0.09$ | $0.78 \pm 0.08$ |
| WBA | $0.39 \pm 0.15$ | $0.74 \pm 0.12$ | $0.34 \pm 0.08$ | $0.78 \pm 0.07$ |
| WBA + masking | $0.39 \pm 0.14$ | $0.74 \pm 0.12$ | $0.37 \pm 0.09$ | $0.77 \pm 0.08$ |
| ZB | $0.44 \pm 0.15$ | $0.71 \pm 0.16$ | $0.51 \pm 0.07$ | $0.65 \pm 0.07$ |
| ZB + masking | $0.43 \pm 0.16$ | $0.69 \pm 0.19$ | $0.54 \pm 0.07$ | $0.63 \pm 0.07$ |
| ZBA | $0.43 \pm 0.16$ | $0.70 \pm 0.15$ | $0.53 \pm 0.06$ | $0.64 \pm 0.05$ |
| ZBA + masking | $0.43 \pm 0.19$ | $0.71 \pm 0.17$ | $0.54 \pm 0.06$ | $0.62 \pm 0.06$ |

Our findings for the baseline aligned with (Zhang et al., 2022) that showed low DSC of 0.25 for the nnU-Net. We observed high performance variability among the cases with lower tumor burden. Segmenting NFs by anatomical zones enhanced accuracy, particularly for the test cases with greater tumor burden (20% increase of DSC). Employing anatomy masks showed only a marginal improvement. We also evaluated the average number of user interactions required to reach a DSC of 0.6, starting from the pre-segmentation masks. When employing the ZBA masks, we observed a decrease in the average number of interactions by 30% (59 interactions), compared to the baseline (80 interactions).

## 4. Conclusion

Our study introduced an anatomy-informed pipeline for NF segmentation in WB-MRI, enhancing preliminary delineation accuracy. By incorporating anatomical information and segmenting based on the specific body zones, we achieved a substantial improvement in the Dice similarity coefficient, highlighting the method's efficacy. Future efforts will focus on integrating this pipeline into a semi-automated workflow to streamline NF segmentation.

## Acknowledgments

This work was supported by DFG grant (DFG SPP 2177), project number 515277218.

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
