# OpenReview forum: "Enhancing Neurofibroma Segmentation in Whole-Body MRI: Leveraging an Anatomy-Informed Approach"
_MIDL.io/2024/Short_Papers — MIDL 2024 Short Papers_

### Official Review · Reviewer_bFvT · 2024-04-23

**Confidence:** 5
**Final Rating:** 3.5

**Review:**

The paper proposed an idea of leveraging anatomical information to improve neurofibroma segmentation in Whole-Body MRI. The topic of automatic neurofibroma segmentation in whole-body MRI is relatively new. However, the proposed method utilized a combination of existing methods - the TotalSegmentator tool and an unsupervised domain adaptation method - to divide the whole body into several regions. Then, a Unet-based segmentation network with additional anatomical information was trained to perform the segmentation. Experiments showed that using anatomical information improved segmentation performance. The main contribution of this paper is the dataset, however, it is unclear whether the author will make the dataset publicly available or not.

Here are detailed comments:

(1) I would suggest to improve the presentation of Fig.1, especially (1) provide a brief description about the pipeline; (2) provide the legend for the segmentation results in the “NFs segmentation” figure

(2) Can you make this sentence more clear  “We observed high performance variability among the cases, with lower variability for the test set featuring a higher tumor load.”? Do you mean the validation set has more variability while the testing set has less variability because of the higher tumor burden?

(3) Please explain what the clinically acceptable DSC or VOE values are for this application? How can your method achieve this goal?

---

### Decision · Program_Chairs · 2024-04-26

Accept